# Glycemic Index, Glycemic Load and Dyslipidemia in Adolescents from Chiapas, Mexico

**DOI:** 10.3390/nu16101483

**Published:** 2024-05-14

**Authors:** Itandehui Castro-Quezada, Pilar Elena Núñez-Ortega, Elena Flores-Guillén, Rosario García-Miranda, César Antonio Irecta-Nájera, Roberto Solís-Hernández, Christian Medina-Gómez, Héctor Ochoa-Díaz-López

**Affiliations:** 1Health Department, El Colegio de la Frontera Sur, Villahermosa 86280, Mexico; itandehui.castro@ecosur.mx (I.C.-Q.); rgarcia@ecosur.mx (R.G.-M.); cirecta@ecosur.mx (C.A.I.-N.); 2Health Department, El Colegio de la Frontera Sur, San Cristóbal de las Casas 29290, Mexico; penunez@guest.ecosur.mx (P.E.N.-O.); rsolis@ecosur.mx (R.S.-H.); christian.medina@guest.ecosur.mx (C.M.-G.); 3Faculty of Nutrition and Food Science, University of Science and Arts of Chiapas, Tuxtla Gutiérrez 29039, Mexico; elena.flores@unicach.mx

**Keywords:** glycemic load, glycemic index, dyslipidemia, adolescents, Chiapas, Mexico

## Abstract

Cardiovascular disease risk throughout the life course is increased by abnormal blood lipid levels in youth. The dietary glycemic index (GI) and glycemic load (GL) during adolescence might be related to abnormal blood lipids. This study aimed to analyze the association between dietary GI, GL and dyslipidemia in adolescents from two marginalized regions of Chiapas, Mexico. A cross-sectional study was conducted with 213 adolescents. Food intake was assessed using 24 h recalls. The association between dyslipidemia and dietary GI or GL was tested by using logistic regression models. Low HDL-c was the most prevalent risk factor (47.4%), followed by hypertriglyceridemia (25.4%). In this population, overall dietary GI was not associated with dyslipidemia. A high dietary GL was associated with 2.39 higher odds of low HDL-c (95% CI: 1.21–4.74) when compared to low GL. Female adolescents with high dietary GL had 3.20 higher odds of hypertriglyceridemia (95% CI: 1.03–9.88), whereas no association was found for males. No associations were observed between overall dietary GL and total cholesterol or LDL-c. In adolescents from urban and rural communities in Chiapas, a high dietary GL was associated with a detrimental effect on HDL-c. In female adolescents, high GL was associated with hypertriglyceridemia.

## 1. Introduction

Cardiovascular diseases are a major cause of death in Mexico. In 2022, among the total deaths caused by heart diseases (200,023 cases), ischemic heart disease represented 76.5%, followed by hypertensive diseases with 14.2% [1]. Similarly, in Chiapas, heart diseases are the main cause of death in men and women, and the incidence of hypertension, coronary heart disease and stroke was 16,118 cases [2]. Most cardiovascular events occur in adulthood; however, evidence has indicated that atherosclerotic lesions begin in childhood and adolescence. The progression of cardiovascular diseases is related to the presence and severity of different risk factors, such as alterations in serum lipid levels and the concentration of lipoproteins in the blood [3]. For instance, high total cholesterol in childhood is a predictor of fatal cardiovascular events in adulthood [4]. Likewise, high-density lipoprotein (HDL-c), low-density lipoprotein (LDL-c) and hypertriglyceridemia in adolescence have been associated with carotid and coronary arteries lesions, considered markers of atherosclerosis [5,6].

The prevalence of dyslipidemias in Mexican adolescents varies according to geographic region and ethnic group. For example, a study conducted on Mexican children and adolescents from North and Central Mexico (Sonora, Puebla and Mexico City) found that the prevalence of dyslipidemia was higher in urban areas (around 76%) and that there was a lower prevalence in certain ethnic groups from the North (Yaquis and Seris) [7]. In adolescents from the northern region, abnormal serum lipid levels were found in 49% [8], while in Mexico City, 25% of adolescents had at least one dyslipidemia [9]. In southern Mexico, Velazco-Martínez et al. [10] found in 259 adolescents from Chiapas that hypercholesterolemia was the most prevalent disorder (26%), followed by hypertriglyceridemia (10%) and elevated LDL-c (7%). In the *Los Altos* and *Selva* regions of Chiapas, low HDL-c prevalence was higher among adolescents whose mothers spoke Mayan than those whose mothers spoke Spanish [11]. Therefore, it is vital to identify modifiable lifestyle factors such as diet and physical activity to control and generate strategies to prevent and manage cardiovascular risk factors in vulnerable population groups.

Dietary recommendations for reducing cardiovascular risk in adolescents suggest including carbohydrates in a range of 50% to 55% of total calories in the diet [12]. However, carbohydrates in foods have different responses in terms of postprandial glucose levels [13]. To measure such variability, the glycemic index (GI) was developed as a tool for diabetes control [14], and it has been used as a valuable measure of the biological quality of carbohydrates in foods [15]. The GI represents the absorption rate of 50 g of carbohydrates from a certain food, expressed as a percentage of the response in the same subject to 50 g of anhydrous glucose or a reference food [14]. The concept of glycemic load (GL) incorporates both the quality and the quantity of digestible carbohydrates [16]. Quite recently, an investigation into the role of dietary GI and GL has been extended to a study of multiple risk factors. For instance, a meta-analysis of nine clinical trials in children and adolescents showed a significant decrease in triglyceride concentrations and the insulin resistance index in children assigned low-GI diets [17]. Nevertheless, a recent study showed that low-GI/GL diets did not affect cardiometabolic risk factors in children with overweight or obesity [18]. 

However, most studies have been conducted among European or U.S. adolescents, and evidence of the effects of high-GI/GL diets on cardiovascular risk factors in indigenous and mestizo populations from marginalized communities in Chiapas, Mexico, is scarce. 

Our working hypothesis was that high-GI and GL diets are associated with dyslipidemia, and this association could be altered by sex or culturally specific foods in adolescents from Chiapas, Mexico. Thus, the objective of this study was to analyze the magnitude of the association between GI and GL and serum lipid levels in adolescents from the Tzotzil–Tzeltal and Selva regions of Chiapas.

## 2. Materials and Methods

### 2.1. Study Design and Population

A cross-sectional study was conducted as part of a cohort study of newborns in two regions of Chiapas, Mexico (Tzotzil-Tzeltal and Selva). The cohort study’s detailed descriptions were previously published [19,20]. A birth cohort was initiated in 2003, which involved studying 2184 newborns from 3 public hospitals in Chiapas [21]. Follow-up visits were carried out for a subsample of the cohort (303 adolescents) [20]. For the sampling design, the localities of residence of the 2003 study participants were grouped into clusters based on their population size and geographical criteria. Both the urban and rural homes of the 2003 newborn cohort were represented by the thirteen clusters that were chosen [20]. The sample size was obtained using the following formula: *n* = [EDFF × N*p*(1 − *p*)]/[(d2/Z21 − α/2 × (N − 1) + *p* × (1 − *p*)]. Here, N = the population size (2184 newborns), *p* = the hypothesized proportion of abnormal blood lipids (25%) and d = the confidence limit (5%). According to these data, the estimated sample size was 255 adolescents.

Trained personnel conducted household interviews, previously standardized, to gather sociodemographic, clinical, dietary and anthropometric data and fasting blood samples. The present study was conducted according to the ethical principles stated in the Declaration of Helsinki. All participants and their parents provided written informed consent/assent. The Research Ethics Committee of El Colegio de la Frontera Sur approved the research protocol (CEI-O- 076/16).

Missing values of total cholesterol (*n* = 47) and triglyceride levels (*n* = 1) and missing data on covariates (*n* = 7) were excluded. Goldberg’s cut-off [22,23] was used to identify implausible reporters of energy intake. The Institute of Medicine’s predictive equations were used to estimate the basal metabolic rate (BMR) [24]. We compared the energy intake reported by participants as a multiple of the BMR using 95% confidence limits [22]. Implausible reporters of energy intake (*n* = 35) were excluded from the analysis. Thus, the final sample was 213 adolescents.

### 2.2. Dietary Assessment

Food intake was assessed by trained dietitians using 24 h recalls. During the interviews, the investigators estimated the portion sizes consumed based on the standard volumes and household measures shown to participants [19]. A nutrient database was compiled by using the Nutrient Composition Tables for Mexican foods [25], food composition tables compiled by the National Institute of Public Health of Mexico [26] and the USDA food composition databases [27]. Energy (kcal/day) and nutrient intakes (g/day) were calculated as the sum of food intake (g/day) multiplied per kcal or grams of the nutrient content of the food (g/g) with a Stata program (Version 17.0, StataCorp, College Station, TX, USA). specifically designed for the foods consumed by the study population.

To estimate the overall dietary GI and GL, a GI value was assigned to each food item reported in the 24 h recalls using a standardized protocol [28]. GI and GL data were obtained from published studies of Mexican foods [29,30] and the International Tables of GI values [31]. Dietary GI was calculated using the following formula [32]: Ʃ [GI × amount of available carbohydrates of each food (g)]/Ʃ [Amount of available carbohydrates of each food (g)]. Dietary GL was estimated as follows: Ʃ [GI × Amount of available carbohydrates of each food (g)]/100. The contribution of foods was evaluated by comparing the GI value or GL intake of food items to the overall dietary GI and GL of the study population.

Total carbohydrates, protein, fat, monounsaturated fatty acids (MUFAs), polyunsaturated fatty acids (PUFAs), saturated fatty acids (SFAs), dietary fiber, total sugars and dietary GL were adjusted for total energy intake using the residuals method [33].

### 2.3. Biochemical Parameters

During the interviews, appointments were scheduled to obtain blood samples from the antecubital area, and they were collected in tubes without anticoagulant after a fasting period of 12 h. The samples were transported in a cold chain to the ECOSUR Health Laboratory, where they were immediately analyzed. Total cholesterol, HDL-c, LDL-c and triglycerides were determined by an enzymatic photometric assay (DiaSys Diagnostic Systems GmbH, Holzheim, Germany) in an automated analyzer (Selectra E, Vital Scientific N.V., Dieren, The Netherlands). 

We categorized blood lipid and lipoprotein concentrations using the American College of Cardiology/American Heart Association Guidelines [34]. The following cut-off points were considered: high total cholesterol levels ≥200 mg/dL; low HDL-c levels <40 mg/dL; and elevated triglyceride concentration ≥130 mg/dL. For LDL-c, we included borderline and high values (≥110 mg/dL) since high LDL-c levels (≥130 mg/dL) were found in only 2 adolescents.

### 2.4. Covariates

We used specific questionnaires to assess sociodemographic characteristics, medical history, and lifestyle habits [20]. The questionnaire included the adolescent’s age, sex, mother’s language (Spanish or indigenous), education of the mother (years) and family history of diabetes, cardiovascular diseases or obesity. Mothers’ schooling data were classified into three categories, considering their last degree: (1) illiterate; (2) elementary school; and (3) middle school, high school or bachelor’s degree.

Anthropometric measurements were taken wearing light clothes and barefoot, according to the International Society for the Advancement of Kinanthropometry procedures [35]. Weight (kg) was measured using electronic scales with a precision of ±0.1 kg (TANITA, Tokyo, Japan). A stadiometer was used to measure standing height (cm) with a precision of ±1 mm (Seca, Hamburg, Germany).

Body mass index (BMI) was calculated as weight divided by height squared (kg/m^2^). We estimated BMI for age Z-scores using the World Health Organization’s (WHO) AnthroPlus software [36]. For this investigation, weight status was classified into two categories based on BMI for age Z-scores or standard deviations (SD) as follows: underweight or normal weight (≤+1 SD) and overweight or obesity (>+1 SD). Body fat percentage (%BF) was estimated with a bioelectrical impedance analysis technique using body composition analyzers (TANITA, Japan). Body fat excess was considered ≥25% BF for males and ≥30% BF for females [37]. Waist circumference was also assessed using a measuring tape with ±1 mm precision (Seca, Germany). Cut-off points for abdominal obesity were defined according to the International Diabetes Federation (waist circumference ≥90 cm for males and ≥80 cm for females) [38].

### 2.5. Statistical Analysis

We categorized dietary GI and energy-adjusted dietary GL into tertiles. Means and standard deviations (SDs), medians and interquartile ranges (IQRs) or percentages were calculated to describe the sample characteristics according to the categories of dietary GI and energy-adjusted dietary GL. To compare characteristics across the dietary GI and GL categories, we used ANOVA or Kruskal–Wallis tests for continuous variables and chi-squared tests for categorical variables.

The association between dyslipidemia and the categories of dietary GI or energy-adjusted dietary GL was tested using multivariable logistic regression models. A purposeful selection method was used for model building [39,40]. Firstly, crude models were fitted to assess the association between variables and outcomes. The variables identified as potential confounders in the first step were included in the multivariable model using forward selection. Covariates with a *p*-value < 0.05 in the Wald test were included in the model. For collinear variables, predictors with the lowest *p*-values were selected. Models were compared using the likelihood ratio test to assess the fit of the model with the additional variables. This process was applied for sociodemographic variables (model 1), anthropometric variables (model 2) and total energy intake and nutrient intake variables (model 3). Finally, the Hosmer–Lemeshow test was used to measure the goodness of fit for the selected multivariable logistic regression models. We also performed tests for linear trends by using the median values of dietary GI or energy-adjusted dietary GL in each category and introduced these variables as a predictor in the multivariable models. As a secondary analysis, the link between the GI or GL of the top food contributors and abnormal blood lipid levels was examined. The models were adjusted for the same variables identified in the main analyses. We assessed potential interactions between dietary GI or GL and sex, geographic area, ethnicity and weight status (underweight or normal vs. overweight or obesity) by introducing the product terms in multivariable models and considered *p* < 0.05 statistically significant in the likelihood ratio test.

## 3. Results

The study sample comprised 213 subjects, of which 106 (49.8%) were female adolescents. The median dietary GI was 51.4 units, and the average dietary GL was 168.8 units per 100 g.

From the foods consumed in this population group, maize tortillas were the major contributor to the total dietary GI and GL (38.7% and 39.6%, respectively) (Appendix A). Also, other food sources of dietary GL and GL were traditional baked goods (*pan dulce*), rice, sugar, cookies, white bread and pasta. Regarding beverages, sweetened soft drinks, lemonade or fruit water with sugar, and juices (industrialized) were among the top ten sources of dietary GI and GL.

Table 1 shows the sociodemographic and health characteristics and dietary intakes of the population according to dietary GI categories. The adolescents in the upper category of dietary GI were younger and lived in urban areas, and their mothers were more likely to have had elementary schooling and Spanish as their primary language when compared to those in the lowest dietary GI category. Regarding health and anthropometric characteristics, adolescents with a high dietary GI were more likely to have a family history of diabetes or CVD and a higher body weight and body fat percentage when compared to those with a low GI diet.

We found differences in dietary intakes across the categories of dietary GI (Table 1). Subjects in the top category (T3) of dietary GI, compared to those in the bottom category (T1), had a lower intake of total carbohydrates, protein and dietary fiber. On the contrary, adolescents with a high dietary GI consumed higher amounts of total fat and total sugars than those with a low dietary GI.

The adolescents with a high dietary GL (Table 2) were predominantly boys and lived mainly in urban areas, and their mothers were more likely to speak an indigenous (Mayan) language. We also observed a non-significant upward trend in the prevalence of overweight/obesity and body fat excess as the GL intake increased. Regarding nutrient intakes, the subjects with a high dietary GL consumed more carbohydrates, dietary fiber and sugars but had a lower protein and total fat intake compared to those with a low dietary GL.

In our sample, low HDL-c levels were the most prevalent risk factor (47.4%), followed by high levels of triglycerides (25.4%) and high total cholesterol (4.2%). Since the prevalence of high LDL-c was low (0.9%), the cut-off of ≥110 mg/dL was used for the analysis (prevalence of borderline and high LDL-c levels = 5.2%). The prevalence of abnormalities in blood lipid levels by dietary GI and dietary GL categories is presented in Table 3. We found a higher prevalence for HDL-c <4 0 mg/dL in adolescents with a high dietary GL (*p* = 0.025). No other statistically significant differences were found for blood lipids across the GI or GL categories. 

Table 4 presents the adjusted odds ratio and 95% CI estimates for the association between abnormal blood lipids and dietary GI. The probability of low HDL-c levels was higher in adolescents with a moderate dietary GI than those with a low GI; nevertheless, such an association was borderline significant, and these results must be confirmed in further studies.

Table 5 shows the ORs and 95% CIs for abnormal blood lipids according to dietary GL categories. In the model adjusted for sex, adolescents with a high dietary GL had higher odds of low HDL-c than adolescents with a low dietary GL (*p for trend* = 0.012). This association remained statistically significant after adjustment for the mothers’ years of education and energy-adjusted dietary fiber intake (OR: 2.19; 95% CI: 1.08–4.42; *p-trend* = 0.029).

Regarding a high level of triglycerides, we found a statistically significant interaction by sex (*p* for interaction < 0.05). Higher odds for hypertriglyceridemia were found for female adolescents with a high dietary GL, whereas no significant association between triglycerides and dietary GL was found for male adolescents. 

In the secondary analysis (Appendix A), we found that the GI of tortillas was associated with lower odds of abnormal total cholesterol and lower odds of borderline and high LDL-c (Appendix A). The GI of cookies was associated with a higher probability of abnormal total cholesterol (OR: 1.03; 95%CI: 1.00–1.06) and hypertriglyceridemia (OR: 1.02; 95%CI: 1.00–1.03).

Regarding the quality and quantity of the carbohydrates in food items and their association with blood lipids (Appendix A), the GL of cookies was significantly associated with high triglyceride levels (OR: 1.03; 95% CI: 1.01–1.06). No significant associations were observed between the GL of the rest of the food items and dyslipidemia. Also, no significant interactions were found between sex and the individual food components.

## 4. Discussion

In the present investigation, conducted with 213 adolescents from scarcely studied and marginalized communities in Chiapas, the overall dietary GI was not significantly associated with abnormal blood lipids. The results revealed that, in the total sample, a high dietary GL was associated with higher odds of low HDL-c but not with total cholesterol or LDL-c. Similarly, in a study conducted on children (8–10 years) from Canada, a high dietary GL was prospectively associated with a decreased HDL-c concentration after 2 years. Also, the authors found that a high dietary GL increased the BMI Z-score, fat mass and triglyceride levels. Furthermore, a mediation by BMI Z-score was found for GL-HDL-c and GL-triglyceride associations [41].

In another study with U.S. adolescents and young adults (11–25 years), an inverse correlation between glycemic load and HDL cholesterol was identified [42]. In our study, dietary GL was negatively correlated with protein intake (−0.41, *p* < 0.001), fat intake (−0.66, *p* < 0.001) and saturated fat intake (−0.51, *p* < 0.001). This finding was also observed by Slyper et al. [42], which reflects an increase in the quantity of highly glycemic carbohydrates consumed at the expense of decreased dietary fat and protein intakes. Previous evidence has demonstrated that diets low in total fat, saturated fat and cholesterol decrease the HDL ApoA-I secretion rate [43] and thus reduce HDL-c levels. In our analyses, the adjustment for protein or fat intake did not change the magnitude or direction of the results; thus, we could assume there is a direct effect of dietary GL on HDL-c.

In our sample, female adolescents with a high dietary GL had higher odds of hypertriglyceridemia than those with a low dietary GL. However, no effect was observed in male adolescents, even though the GL values across the categories were similar among the sexes. This finding suggests that the female adolescents in this population might be more susceptible to high-GL diets. A possible explanation is that pubertal status may be involved. Although such data were not available for our analyses, previous evidence has demonstrated that blood lipid levels can change depending on pubertal stage, sex and race [44]. Sex differences in body composition could also explain the latter finding. In a previous study conducted in the same population, female adolescents had a significantly higher prevalence of abdominal obesity than male adolescents [12]. High triglyceride levels are common in children and adolescents with obesity [45]. Nevertheless, the logistic regression models that were adjusted for body fat and included abdominal obesity as a covariate did not modify the magnitude or direction of the effect. Sex differences in physical activity might be involved in the association between GL and triglycerides. The results from ENSANUT 2018 showed that a lower proportion of girls met the physical activity recommendations (41.3%) compared to the boys (55.5%) [46]. 

Findings from a meta-analysis of nine randomized clinical trials showed that low-GI/GL diets decreased serum triglycerides as compared to high-GI/GL diets in both male and female children and adolescents, but no other blood lipids were reduced [17]. Within the first 2 h after a high-GI/GL meal, hyperglycemia and hyperinsulinemia stimulate the uptake of nutrients through insulin-responsive tissues, glycogenesis and lipogenesis. From 2 to 4 h after food intake, the blood glucose concentration may decrease into the hypoglycemic range. In the late postprandial period (4 to 6 h after the meal), euglycemia is restored by hormone responses that activate gluconeogenic and glycogenolytic pathways, raising free fatty acid levels to levels higher than those documented after a low-GI/GL meal [47]. Therefore, prospective studies on low-GI/GL diets conducted in Mexican adolescents are needed to confirm our results regarding the sex-specific association.

In adults, Culberson et al. [48] reported that a high GL was associated with low HDL-cholesterol levels but not with other components of metabolic syndrome. Another study conducted on adults and the elderly population in Brazil found that the GI and GL were also associated with a low HDL-c concentration [49]. Such evidence is not consistent; for instance, two meta-analyses from randomized clinical trials with adults showed that low-GI diets reduced total cholesterol and LDL-c, but no effect was found on HDL-cholesterol or triglycerides [50,51]. On the contrary, a recent meta-analysis showed no beneficial effect of low-GI diets on blood lipids [52].

In our study, no statistically significant associations were found between overall dietary GI and abnormalities in blood lipids. Research on school-age children in China showed that a high-GI diet was associated with higher triglyceride levels and lower HDL-c concentrations, and a moderate GL was associated with higher serum LDL-c levels [53]. The disparities among studies may be explained by differences in food intake and dietary habits. The study from China was carried out in Guangdong, and as the authors explained, most carbohydrates in the children’s diet were derived from white rice (a high-GI food); in contrast, in our sample, most carbohydrates were derived from maize products such as “tortillas”. In the Guangdong study, the upper GI category comprised 66.2 to 100.0 units while in our study the range for the highest category was 53.0 to 62.7 units of dietary GI. Therefore, it is possible that our sample did not reach the highest values of dietary GI consumption that could be associated with abnormal blood lipids in adolescents.

In the secondary analyses, the GI of cookies was directly associated with abnormal levels of total cholesterol and hypertriglyceridemia. In Mexico, cookies are one of the main sources of saturated fatty acids [54], high-fructose corn syrup, and/or sugar, a nutrient profile linked to dyslipidemia [55,56]. High-GI foods rich in sugars increase small and dense LDL-c by increasing plasma triglyceride levels [57]. On the contrary, in our study, the GI of tortillas was inversely associated with abnormal total cholesterol and LDL-c levels. Tortillas have a low GI, and their fiber content might be involved in reduced dietary cholesterol absorption and increased bile acid excretion [58]. For instance, a study conducted on animals showed that the group fed tortillas had lower LDL-c levels after 28 weeks [59]. Nevertheless, these results should be taken with caution since more evidence is necessary to confirm the association between the GI of certain foods, such as tortillas, and dyslipidemia in youth populations.

A limitation of the present study is that the estimation of each individual’s energy and nutrient intakes was carried out with one 24 h diet recall interview. Previous studies have shown that energy intake is underreported with this method [60]. Also, the 24 h diet recall might not appraise intra-individual variations in food intake and could be biased by memory or likeability. Nevertheless, in our study, the interview was administered to adolescents and their mothers by trained nutritionists using plastic food models and tablets for data collection and real-time codification. Furthermore, we excluded from the analysis implausible reporters of energy intake [19]. The sample size of our study and the limited number of cases of abnormal total cholesterol and LDL-c might have limited the statistical power of our analysis, hindering it from showing a significant association with those outcomes. However, the estimated statistical power for the models adjusted for HDL-c and dietary GL was 80%. The lack of information on medications that could affect blood lipid metabolism was another limitation. However, none of the participants mentioned a previous diagnosis of dyslipidemia. Finally, a constraint in our study is that a reliable measure of the physical activity of adolescents was not available to use as a potential confounder in the logistic regression models.

One of the strengths of this investigation is the study population; this sample of adolescents from southern Mexico included marginalized mestizo and indigenous communities where no previous studies have been conducted to assess the association between dietary GI and GL and abnormal lipid levels. Another strength was that, despite the lack of certain GI values for some Mexican foods, we used a validated protocol to assign most of the GI values to food items using direct assignment and, to a lesser extent, the values of similar foods in terms of nutrient composition [19,28].

In conclusion, overall dietary GI was not associated with dyslipidemia in adolescents from Chiapas, Mexico. In the total sample, high dietary GL was associated with low HDL-c levels. Female adolescents in the top dietary GL category had higher odds of hypertriglyceridemia than those in the bottom category of dietary GL. No associations were observed between overall dietary GL and total cholesterol or LDL-c. The results suggest that there is a detrimental effect of high-GL diets on cardiovascular risk factors in vulnerable populations living in marginalized communities in the south of Mexico.

## Figures and Tables

**Table 1 nutrients-16-01483-t001:** Sociodemographic, health and anthropometric characteristics and energy and nutrient intakes according to dietary glycemic index (GI) categories in adolescents from Chiapas, Mexico.

Variables ^a^	Dietary GI				*p*-Value ^b^
	*T*1	*T*2	*T*3	Total	
	*n* = 71	*n* = 71	*n* = 71	*n* = 213	
Dietary GI (units) ^c^	46.5 (44.5–47.7)	51.4 (50.1–52.2)	56.2 (54.4–58.0)	51.4 (47.7–54.4)	<0.001 ^d^
Sociodemographic characteristics					
Sex (% females)	42.3	52.1	54.9	49.8	0.284
Age (years) ^c^	14.2 (14.0–14.4)	14.1 (14.0–14.3)	14.0 (13.9–14.2)	14.1 (14.0–14.3)	<0.001 ^d^
Geographic area (%)					0.002
Urban	63.4	76.1	88.7	76.1	
Rural	36.6	23.9	11.3	23.9	
Mother’s schooling level (%)					0.005
Illiterate	52.1	33.8	23.9	36.6	
Elementary school	39.4	45.1	59.2	47.9	
Middle school, high School or bachelor’s degree	8.5	21.1	16.9	15.5	
Mother’s language (%)					<0.001
Spanish	39.4	53.5	73.2	55.4	
Indigenous (Mayan)	60.6	46.5	26.8	44.6	
Health and anthropometric characteristics					
Family history of diabetes (%)	35.2	42.3	57.8	45.1	0.022
Family history of obesity (%)	19.7	25.4	25.4	23.5	0.658
Family history of CVD (%)	36.6	36.6	57.8	43.7	0.014
Weight (kg) ^c^	46.6 (41.8–51.0)	48.8 (43.9–54.1)	50.7 (45.9–56.3)	48.8 (44.0–53.4)	0.004 ^d^
Weight status (%)					0.313
Underweight/normal weight	78.9	71.8	67.6	72.8	
Overweight/obesity	21.1	28.2	32.4	27.2	
Waist circumference (cm) ^c^	70.5 (67.5–76.0)	72.8 (69.0–77.0)	72.5 (68.0–77.8)	72.0 (68.0–77.0)	0.094 ^d^
Abdominal obesity (%)	9.9	16.9	15.5	14.1	0.443
% Body fat ^c^	18.0 (12.7–25.1)	23.4 (15.4–29.0)	25.9 (18.9–29.8)	22.9 (15.1–28.4)	0.001 ^d^
Body fat excess (%)	12.7	21.1	31	21.6	0.03
Energy and nutrient intakes					
Energy intake (kcal per day) ^c^	2107 (1803–2480)	2088 (1642–2663)	2132 (1860–2586)	2125 (1775–2553)	0.814 ^d^
Total carbohydrates (g per day) ^e^	336.1 (57.6)	340.9 (40.0)	320.9 (48.1)	332.6 (49.6)	0.042
Total carbohydrates (% Energy per day)	61.2 (11.1)	62.3 (7.7)	58.4 (8.9)	60.7 (9.4)	0.041
Protein (g per day) ^ce^	71.4 (64.0–88.0)	67.6 (59.8–83.1)	67.0 (49.5–81.1)	69.4 (58.7–82.5)	0.007 ^d^
Protein (% Energy per day) ^c^	13.2 (11.6–15.5)	12.2 (10.6–14.6)	12.1 (9.1–14.3)	12.6 (10.6–14.6)	0.005 ^d^
Total fat (g per day) ^ce^	65.8 (44.8–84.6)	61.4 (51.9–76.6)	73.4 (60.9–84.3)	68.0 (54.2–81.8)	0.007 ^d^
Total fat (% Energy per day)	26.0 (9.9)	25.9 (7.0)	30.7 (8.8)	27.5 (8.9)	0.001
Dietary fiber (g per day) ^ce^	32.6 (23.9–42.2)	29.9 (22.1–35.0)	22.2 (16.8–26.4)	26.5 (20.1–34.1)	<0.001 ^d^
Total sugars (g per day) ^ce^	72.0 (51.8–91.3)	79.2 (55.3–98.1)	84.8 (58.2–126.2)	77.4 (54.4–102.1)	0.044 ^d^

^a^ Means and standard deviation (SD) or percentages are shown. ^b^ Differences between dietary glycemic index categories were tested using ANOVA (continuous variables) or chi-square tests (categorical variables). ^c^ Medians (Interquartile range). ^d^ Kruskal–Wallis test. ^e^ Energy-adjusted (residuals method).

**Table 2 nutrients-16-01483-t002:** Sample characteristics and energy and nutrient intakes according to energy-adjusted dietary glycemic load (GL) categories in adolescents from Chiapas, Mexico.

Variables ^a^	Energy-Adjusted Dietary GL				*p*-Value ^b^
	*T*1	*T*2	*T*3	Total	
	*n* = 71	*n* = 71	*n* = 71	*n* = 213	
Dietary GL (g/day) ^c^	137.3 (17.6)	170.7 (7.5)	198.5 (13.9)	168.8 (28.5)	<0.001
Sociodemographic characteristics					
Sex (% female)	54.9	56.3	38	49.8	0.052
Age (years) ^d^	14.1 (14.0–14.2)	14.1 (13.9–14.3)	14.2 (13.9–14.3)	14.1 (14.0–14.3)	0.784 ^e^
Geographic area (%)					0.009
Urban	88.7	70.4	69	76.1	
Rural	11.3	29.6	31	23.9	
Mother’s schooling level (%)					0.113
Illiterate	28.2	42.3	39.4	36.6	
Elementary school	47.9	45.1	50.7	47.9	
Middle school, high school or bachelor’s degree	23.9	12.7	9.9	15.5	
Mother’s language (%)					0.040
Spanish	66.2	54.9	45.1	55.4	
Indigenous (Mayan)	33.8	45.1	54.9	44.6	
Health and anthropometric characteristics					
Family history of diabetes (%)	54.9	43.7	36.6	45.1	0.087
Family history of obesity (%)	19.7	25.4	25.4	23.5	0.658
Family history of CVD (%)	40.9	43.7	46.5	43.7	0.795
Weight (kg) ^d^	47.1 (42.2–52.8)	48.6 (44.8–53.4)	49.8 (44.0–54.3)	48.8 (44.0–53.4)	0.361 ^e^
Weight status (%)					0.515
Underweight/normal weight	71.8	77.5	69	72.8	
Overweight/obesity	28.2	22.5	31	27.2	
Waist circumference (cm) ^d^	70.0 (67.0–76.0)	72.8 (69.0–77.0)	73.0 (67.5–77.7)	72.0 (68.0–77.0)	0.156 ^e^
Abdominal obesity (%)	12.7	15.5	14.1	14.1	0.89
% Body fat ^d^	23.4 (15.4–29.4)	23.3 (16.7–29.0)	22.1 (14.7–27.4)	22.9 (15.1–28.4)	0.842 ^e^
Body fat excess (%)	21.1	18.3	25.4	21.6	0.59
Energy and nutrient intakes					
Energy intake (kcal/day) ^d^	2187 (1915–2532)	2052 (1700–2516)	2104 (1759–2663)	2125 (1775–2553)	0.309 ^e^
Total carbohydrates (g/day) ^c^	289.2 (40.9)	332.1 (30.3)	376.5 (31.6)	332.6 (49.6)	<0.001
Total carbohydrates (% Energy/day)	52.4 (7.7)	60.7 (6.2)	68.9 (5.9)	60.7 (9.4)	<0.001
Protein (g/day) ^cd^	76.2 (65.2–93.7)	69.3 (59.7–85.5)	62.3 (51.9–71.8)	69.4 (58.7–82.5)	<0.001 ^e^
Protein (% Energy/day) ^d^	14.0 (11.9–17.6)	12.7 (10.6–15.4)	11.3 (9.4–13.0)	12.6 (10.6–14.6)	<0.001 ^e^
Total fat (g/day) ^cd^	83.8 (74.9–90.5)	69.6 (57.5–77.7)	53.3 (42.1–61.4)	68.0 (54.2–81.8)	<0.001 ^e^
Total fat (% Energy/day)	33.9 (8.2)	27.5 (7.1)	21.1 (6.3)	27.5 (8.9)	<0.001
Dietary fiber (g/day) ^cd^	23.9 (18.6–30.5)	27.5 (20.1–34.1)	31.1 (23.0–38.1)	26.5 (20.1–34.1)	0.001 ^e^
Total sugars (g/day) ^cd^	68.7 (47.6–91.4)	77.1 (55.2–114.1)	83.6 (62.1–109.5)	77.4 (54.4–102.1)	0.032 ^e^

^a^ Means and standard deviation (SD) or percentages are shown. ^b^ Differences between categories of energy-adjusted dietary glycemic load were tested using ANOVA (continuous variables) or chi-square tests (categorical variables). ^c^ Energy-adjusted (residuals method). ^d^ Medians (Interquartile range). ^e^ Kruskal–Wallis test.

**Table 3 nutrients-16-01483-t003:** Prevalence of abnormal blood lipids by dietary GI and energy-adjusted dietary GL categories.

Variables ^a^	Dietary GI			*p*-Value ^b^	Energy-Adjusted Dietary GL			*p*-Value ^b^
	*T*1	*T*2	*T*3		*T*1	*T*2	*T*3	
	*n* = 71	*n* = 71	*n* = 71		*n* = 71	*n* = 71	*n* = 71	
Total cholesterol ≥ 200 mg/dL (%)	4.2	1.4	7.0	0.249	4.2	5.6	2.8	0.706
HDL-c < 40 mg/dL (%)	42.3	54.9	45.1	0.283	35.2	49.3	57.8	0.025
LDL-c ≥ 110 mg/dL (%)	4.2	2.8	8.5	0.288	4.23	8.45	2.82	0.288
Triglycerides ≥ 130 mg/dL	28.2	22.5	25.4	0.743	21.1	23.9	31.0	0.380

^a^ Percentages are shown. ^b^ Differences across categories were analyzed using chi-square tests for categorical data.

**Table 4 nutrients-16-01483-t004:** Odds ratios and 95% confidence intervals for abnormal blood lipids by dietary GI categories.

Variables		Dietary GI			*p*-Trend
		*T*1	*T*2	*T*3	
		*n* = 71	*n* = 71	*n* = 71	
	GI Median	46.5	51.4	56.2	
Total cholesterol ≥ 200 mg/dL	Cases	3	1	5	
	Model 1 ^a^	1 (Ref.)	0.28 (0.03–2.84)	1.49 (0.33–6.60)	0.516
	Model 2 ^b^	1 (Ref.)	0.23 (0.02–2.38)	1.08 (0.23–5.16)	0.769
	Model 3 ^c^	1 (Ref.)	0.31 (0.03–3.38)	1.51 (0.28–8.06)	0.510
HDL-c < 40 mg/dL	Cases	30	39	32	
	Model 1 ^a^	1 (Ref.)	1.75 (0.89–3.42)	1.19 (0.61–2.32)	0.612
	Model 2 ^d^	1 (Ref.)	1.91 (0.96–3.80)	1.44 (0.71–2.93)	0.305
	Model 3 ^e^	1 (Ref.)	1.77 (0.88–3.54)	1.23 (0.59–2.56)	0.573
LDL-c ≥ 110 mg/dL	Cases	3	2	6	
	Model 1 ^a^	1 (Ref.)	0.57 (0.09–3.57)	1.79 (0.42–7.61)	0.361
	Model 2 ^b^	1 (Ref.)	0.48 (0.07–3.15)	1.40 (0.31–6.29)	0.545
	Model 3 ^f^	1 (Ref.)	0.80 (0.11–5.88)	2.64 (0.48–14.47)	0.209
Triglycerides ≥ 130 mg/dL	Cases	20	16	18	
	Model 1 ^a^	1 (Ref.)	0.70 (0.32–1.51)	0.81 (0.38–1.71)	0.571
	Model 2 ^b^	1 (Ref.)	0.63 (0.29–1.40)	0.66 (0.30–1.45)	0.295
	Model 3 ^g^	1 (Ref.)	0.75 (0.33–1.70)	1.03 (0.42–2.51)	0.992

^a^ Adjusted for sex. ^b^ Adjusted for sex and body fat percentage (normal vs. high). ^c^ Adjusted for sex, body fat percentage (normal vs. high) and energy-adjusted protein intake (grams per day). ^d^ Adjusted for sex and language of mother (Spanish vs. indigenous). ^e^ Adjusted for sex, language of mother (Spanish vs. indigenous) and protein intake (percentage of total energy intake per day). ^f^ Adjusted for sex, body fat percentage (normal vs. high) and protein intake (percentage of total energy intake per day). ^g^ Adjusted for sex, body fat percentage (normal vs. high) and energy-adjusted dietary fiber intake (grams per day).

**Table 5 nutrients-16-01483-t005:** Odds ratios and 95% confidence intervals for the association between abnormal blood lipids and dietary GL categories.

Variables		Energy-Adjusted Dietary GL			*p*-Trend
		*T*1	*T*2	*T*3	
		*n* = 71	*n* = 71	*n* = 71	
	GL Median	143	171.9	197.1	
Total cholesterol ≥ 200 mg/dL	Cases	3	4	2	
	Model 1 ^a^	1 (Ref.)	1.34 (0.29–6.27)	0.80 (0.13–5.04)	0.870
	Model 2 ^b^	1 (Ref.)	1.53 (0.31–7.46)	0.69 (0.11–4.46)	0.764
	Model 3 ^c^	1 (Ref.)	2.39 (0.41–13.78)	1.54 (0.17–13.58)	0.602
HDL-c < 40 mg/dL	Cases	25	35	41	
	Model 1 ^a^	1 (Ref.)	1.80 (0.92–3.55)	2.39 (1.21–4.74)	0.012
	Model 2 ^d^	1 (Ref.)	1.71 (0.86–3.38)	2.30 (1.16–4.58)	0.017
	Model 3 ^e^	1 (Ref.)	1.66 (0.83–3.30)	2.19 (1.08–4.42)	0.029
LDL-c ≥ 110 mg/dL	Cases	3	6	2	
	Model 1 ^a^	1 (Ref.)	2.09 (0.49–8.83)	0.82 (0.13–5.20)	0.977
	Model 2 ^f^	1 (Ref.)	2.33 (0.52–10.37)	0.63 (0.10–4.15)	0.757
	Model 3 ^g^	1 (Ref.)	4.63 (0.82–26.24)	1.90 (0.20–17.63)	0.422
Triglycerides ≥ 130 mg/dL ^h^	Males (*n* = 107)	*n* = 36	*n* = 36	*n* = 35	
	Median (sex-specific)	142.9	177.0	199.0	
	Cases	9	5	8	
	Model 1 ^i^	1 (Ref.)	0.50 (0.15–1.68)	1.05 (0.34–3.22)	0.888
	Model 2 ^j^	1 (Ref.)	0.50 (0.14–1.74)	0.93 (0.29–3.00)	0.754
	Model 3 ^k^	1 (Ref.)	0.55 (0.16–1.97)	1.25 (0.36–4.36)	0.919
	Females (*n* = 106)	*n* = 36	*n* = 35	*n* = 35	
	Median (sex-specific)	143.3	165.4	192.6	
	Cases	6	11	15	
	Model 1 ^i^	1 (Ref.)	1.97 (0.62–6.25)	3.20 (1.03–9.88)	0.043
	Model 2 ^j^	1 (Ref.)	1.97 (0.61–6.37)	3.18 (1.01–9.99)	0.047
	Model 3 ^k^	1 (Ref.)	3.16 (0.84–11.83)	6.71 (1.56–28.98)	0.011

^a^ Adjusted for sex. ^b^ Adjusted for sex and weight status (underweight or normal vs. overweight or obesity). ^c^ Adjusted for sex, weight status (underweight or normal vs. overweight or obesity) and protein intake (percentage of total energy intake per day). ^d^ Adjusted for sex and mother’s education (<6 years vs. ≥6 years). ^e^ Adjusted for sex, mother’s education (<6 years vs. ≥6 years) and energy-adjusted dietary fiber intake (grams per day). ^f^ Adjusted for sex and body fat percentage (normal vs. high). ^g^ Adjusted for sex, body fat percentage (normal vs. high) and energy-adjusted protein intake (grams per day). ^h^ p for interaction < 0.05. ^i^ Adjusted for mother’s language (Spanish vs. Mayan). ^j^ Adjusted for mother’s language (Spanish vs. Mayan) and body fat percentage (normal/high). ^k^ Adjusted for mother’s language (Spanish vs. Mayan), body fat percentage (normal vs. high) and protein intake (percentage of total energy intake per day).

## Data Availability

The data presented in this study are available on request from the corresponding author.

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
