# Peer review of "Glycemic Index, Glycemic Load and Dyslipidemia in Adolescents from Chiapas, Mexico"

_nutrients, 2024, doi:10.3390/nu16101483_

Round 1
Reviewer 1 Report
Comments and Suggestions for Authors
Dear Redactors,
Thank you very much for the possibility to revise the article “Dietary Glycemic Index, Glycemic Load and Dyslipidemia in Adolescents from Chiapas, Mexico”.
The article is very interesting and well-written.
I have just a few recommends.
In the introduction part, please add more information about epidemiology of cardiovascular diseases.
In methodology section, please add information about medicines and therapies the participants were going under. It is also important to describe the methods of estimating the serving size (was it done by participants by themselves?) and the computer facilities which were used to check the energy and nutrients intake.
Were the blood tests were taken immediately after blood collection or wheter the blood tests were frozen?
In the discussion please describe in more details what are the possible explanations of gender differences in tested markers.
Thanks
Reviewer 2 Report
Comments and Suggestions for Authors
This study aimed to analyze the association between dietary GI, dietary GL and dyslipidemia in adolescents from the Tzot-zil-Tzeltal and Selva regions of Chiapas, Mexico. This is a great study, but requires several changes prior the acceptance for publication.
Abstract:
-To add more numbers in the abstract.
Introduction:
-What is hypothesis of study? to clarify
Methods:
-What is sample size calculus?
Results:
-What is physical activity level of patients? This is important to understand these data.
Discussion:
Is importnt to discussion if female adolescents to ingest the energy drink? In addition, softdrink with seeweters are a concern of this population. Please, to dissuss it.
24h dietary recall is very complicated to establish the real data regarding to habitual food intake (limitation).
Reviewer 3 Report
Comments and Suggestions for Authors
This is an interesting study to examine the relationship between glycemic index, glycemic load and dyslipidemia among young people in Mexico. I have several comments to improve the manuscript.
First, the introduction is a bit jumpy from the current evidence on GI/GL and dyslipidemia to proposing the present study. Especially when meta-analyses are available, what is the research gap doing a similar study in Mexico? Any justification to urge the need of doing a similar study?
For the analysis plan, apart from GI/GL, authors should present the top food/beverages that contribute the most to GI/GL, so that the findings can help to inform dietary guidelines. Food/beverages with high GI/GL can differ by population, and can be culturally specific, so authors can identify those food/beverages first, then as a secondary analysis, analyse the association between individual food/beverage with dyslipidemia.
Comments on the Quality of English LanguageNil
Round 2
Reviewer 3 Report
Comments and Suggestions for Authors
My concerns have been addressed, and have no more further comments.
Comments on the Quality of English LanguageNil.
Author Response
We appreciate Reviewer´s comments, which have been very helpful to improve our manuscript.